# Overcoming Difficulties in Molecular Biological Analysis through a Combination of Genetic Engineering, Genome Editing, and Genome Analysis in Hexaploid *Chrysanthemum morifolium*

**DOI:** 10.3390/plants12132566

**Published:** 2023-07-06

**Authors:** Katsutomo Sasaki, Tsuyoshi Tanaka

**Affiliations:** 1Institute of Vegetable and Floriculture Science, National Agriculture and Food Research Organization (NARO), 2-1 Fujimoto, Tsukuba 305-0852, Ibaraki, Japan; 2Research Center for Advanced Analysis, National Agriculture and Food Research Organization (NARO), 2-1-2 Kannondai, Tsukuba 305-8518, Ibaraki, Japan; tstanaka@affrc.go.jp

**Keywords:** biotechnology, breeding, chrysanthemum, genetic engineering, genome analysis, genome editing, hexaploid, transcriptome analysis

## Abstract

Chrysanthemum is one of the most commercially important ornamental plants globally, of which many new varieties are produced annually. Among these new varieties, many are the result of crossbreeding, while some are the result of mutation breeding. Recent advances in gene and genome sequencing technology have raised expectations about the use of biotechnology and genome breeding to efficiently breed new varieties. However, some features of chrysanthemum complicate molecular biological analysis. For example, chrysanthemum is a hexaploid hyperploid plant with a large genome, while its genome is heterogeneous because of the difficulty of obtaining pure lines due to self-incompatibility. Despite these difficulties, an increased number of reports on transcriptome analysis in chrysanthemum have been published as a result of recent technological advances in gene sequencing, which should deepen our understanding of the properties of these plants. In this review, we discuss recent studies using gene engineering, genome editing, and genome analysis, including transcriptome analysis, to analyze chrysanthemum, as well as the current status of and future prospects for chrysanthemum.

## 1. Introduction

*Chrysanthemum morifolium* (chrysanthemum) is one of the most commercially important ornamental flowers globally. Although the flower (head flower) of chrysanthemum (Chrysanthemumaceae) appears to be a single flower, it is actually a compound flower composed of many small flowers, called florets. Chrysanthemum basically contains two types of florets, which can be clearly distinguished by their shapes. The petal-like small florets located on the outside of the head flower are ray florets, while the small yellow florets at the center are disk florets. Although not seen in all chrysanthemum flowers, a third type of small floret, the longer disk-type florets, which are usually similar in color to the ray florets, are found between the ray and disk florets in anemone-type chrysanthemum. Many chrysanthemum varieties with different flower colors and types have been established [1,2], with numerous new varieties being created every year. Crossbreeding is currently the most popular method of creating new chrysanthemum varieties [3]. However, this approach is laborious, raising the need to use molecular breeding to efficiently breed new varieties. Other attempts to create new varieties by radiation-induced mutation breeding have also been carried out.

Recently, remarkable advances have been achieved in biotechnology and gene/genome sequencing. In this context, attempts are being made to modify chrysanthemum traits via genetic modification, genome editing, and genome breeding, among others. However, these biotechnologies are not as readily applicable to chrysanthemum as they are to other plants because of the higher ploidy (hexaploid) of chrysanthemum. Chrysanthemum also has a very large genome (12.66–25.36 Gbp) [4] and typically 54 chromosomes or similar (2n = 6x = 54) [5]. However, chrysanthemum can widely vary in some of its traits, such as exhibiting various ploidy levels and aneuploidy (with the chromosome number ranging from 47 to 67) [6,7,8]. Moreover, chrysanthemum is essentially self-incompatible [9], severely complicating the possibility of obtaining pure lines. The genome of chrysanthemum is also not genetically fixed, unlike those of model plants such as rice, tomato, and *Arabidopsis*, so the characteristics of parental lines (e.g., varieties, T_0_ transgenic plants, and T_0_ genome-edited plants) with respect to floral traits and cultivar traits, among others, are not fully reproduced in the next generation derived from their seeds. Therefore, it is virtually impossible to develop F_1_ varieties in chrysanthemum, in contrast to commercially available varieties of other floral commodities. In this context, chrysanthemum plants and cultivars of the same lineage are vegetatively propagated through cuttings to increase the number of individuals, including varieties. Considering the large genome and genomic heterogeneity, it is still difficult to establish genomic information on chrysanthemum and the whole-genome sequence of chrysanthemum has still not been established.

This review discusses the use of biotechnology in chrysanthemum, the development of genomic information on these plants, and the future prospects for using biotechnology in chrysanthemum, considering the aforementioned characteristics of these plants.

## 2. Conventional Breeding in Chrysanthemum: Crossbreeding and Mutagen Breeding for the Generation of New Varieties

New varieties of chrysanthemum have been developed for commercial use, primarily through crossbreeding [3]. In chrysanthemum crossbreeding, as in the breeding of other plants, the pollen and seed parents harboring the target traits are first prepared. In the ray of chrysanthemum florets, there is no stamen, but only a pistil is present. In tubular flowers, both stamens and pistils are present. Since the pistils of disk florets are more fertile than those of ray florets, the disk florets are used for pollination. Prior to cross-pollination, the flower heads of the seed parents should be bagged to avoid unintended pollination by insects. To facilitate cross-pollination, the ray florets are cut off with scissors to expose the disk florets at the time of cross-pollination. Chrysanthemum pollen is short-lived, lasting only a day or two, and therefore, fresh pollen should be used. Compared to the pollen, the pistil remains viable for pollination for a longer period of time, and after the stigma opens it can be pollinated with a cotton swab. The seeds are fully ripe after approximately two months, depending on the season. Once a line with the desired traits has been obtained from the crossbred seeds, propagation of the varieties obtained can be performed by vegetative propagation through cuttings.

Branching mutants have also been developed from already obtained varieties. Another method for variety development is breeding using irradiation-induced mutants. Some of these irradiated mutants have actually been developed for commercial use. As explained above, by their nature, commercial varieties of chrysanthemum cannot be propagated and distributed via their seeds. They are instead propagated by vegetative propagation from cuttings. Types of radiation that induce mutation include gamma rays, X-rays, and heavy-ion-beam irradiation, which have been used to produce many mutants with altered flower color via radiation breeding [10,11,12,13,14]. Among the multiple available options of irradiation types for creating mutations, heavy-ion-beam irradiation has been used for mutagen breeding in a particularly wide variety of ornamental flowers, including chrysanthemum [10,14]. This approach is more efficient for introducing mutations than other radiation types because the linear energy transfer (LET, the energy transferred per unit length, keV mm^−1^) of heavy-ion beams is considerably higher than those of gamma rays and X-rays [15,16]. In general, irradiation with higher LET exerts stronger biological effects than that with lower LET. In chrysanthemum, previous studies involving heavy-ion-beam irradiation produced phenotypic changes in flower color [17], induced early flowering at low temperatures [18], and resulted in reductions in the number of axillary buds [19]. In some cases, branch-swapping mutants were obtained by directly irradiating the plant itself with heavy-ion beams, while in others, individuals regenerated from adventitious shoots in a sterile culture. Many varieties have been produced not only by such crosses but also by mutagenesis, but it is very difficult to efficiently introduce mutations only in the target gene during mutagenesis, so luck is essentially required to produce the target trait.

## 3. Development of Biotechnology in *Chrysanthemum* and Objectives of Its Use

### 3.1. Various Biotechnologies in Plants and the Objectives of Their Use

When studying higher plants, it is important to set a research problem and to choose a method to overcome this problem, such as by determining which biotechnological method(s) is most appropriate. In fact, it may be preferable to select a combination of methods to solve the research problem. Biotechnological research on higher plants has been widely conducted, mainly using mutants and transgenic plants in *Arabidopsis* (a model plant for molecular biological studies), to analyze gene functions and to elucidate changes in plant traits. However, in recent years, research topics have shifted from basic research focusing on model plants to the next step, namely, applied research for commercial trait modification of commercial crops and for improving commercial cultivation. Therefore, increasing numbers of plant species are now being treated as targets for analysis, using information accumulated on both model plants and commercially important plant species such as chrysanthemum.

### 3.2. Genetic Engineering in Chrysanthemum

To date, an *Agrobacterium tumefaciens*-based method has mainly been applied for the genetic modification of chrysanthemum [20,21]. However, this approach cannot be used for the genetic modification of some varieties. In such varieties, modified versions of the genetic modification procedure may be required, such as by adjusting the concentrations of antibiotics and plant hormones. This problem is salient in chrysanthemum because of its large number of varieties, but also applies to other ornamental plants as well.

Reports using transgenic chrysanthemum plants are gradually accumulating [2]. However, there is still a lack of information on regulatory expression tools and analytical methods to overcome the difficulties in gene analysis due to the complicated floral structure, high polyploidy, and large genome of chrysanthemum. In chrysanthemum, several points need to be considered when creating targeted transgenic plants, in order to achieve the desired gene expression. To modify the desired trait, it is necessary to express the transgene in the target tissues and organs at the required timing and at the required levels, in terms of both overexpression and gene silencing. For transgene expression and suppression, available sequences for the vector specifically include promoters, terminators, or translation promotion factors (Table 1). Other sequences that can potentially be used for modifying the function of transcription factors in plants are repression domains (RDs) and activation domains (ADs) for transcription factors (TFs) (Table 1). Below, we present three examples on improving transgene expression regarding the acquisition of target traits in transgenic chrysanthemum.

### 3.3. An Example of Improving Transgene Expression to Obtain Target Traits in Transgenic Chrysanthemum, Part 1

One example experiment involved an effort to simultaneously suppress the expression of two types of chrysanthemum class C genes, homologs of the *AGAMOUS* (*AG*) gene, *CAG1* and *CAG2*, by RNAi using the CaMV 35S (35S) promoter. We undertook a study to generate transgenic chrysanthemum plants with a petalized phenotype in stamens and pistils by suppressing the function of the class C genes. Since two types of class C genes, *CAG1* and *CAG2*, were identified in chrysanthemum [28], it was assumed that it would be necessary to suppress the function of two types of TFs to acquire the target trait that we required. This is because it was reported that, in antisense transgenic plants using only *CAG1*, one of the class C genes in chrysanthemum, complete alteration of stamens and pistils did not occur [32]. First, *CAG1* and *CAG2* RNAi vectors were produced using the 35S promoter and introduced into chrysanthemum (Figure 1a). However, the use of the 35S promoter did not alter the floral traits (Figure 1b). We then isolated and utilized the chrysanthemum *Actin2* (*CmACT2*) promoter [26] in the RNAi vector, instead of the 35S promoter (Figure 1a), and this led to production of the desired trait, with stamens and pistils being transformed into petaloid organs (Figure 1b) [28]. For the RNAi vector produced in that study, the terminator was also upgraded from HSPT to HSPT878, which is reported to have stronger transcription termination activity [33].

### 3.4. An Example of Improving Transgene Expression to Obtain Target Traits in Transgenic Chrysanthemum, Part 2

The class C genes encode TFs for the ABC model genes. Powerful tools have been developed to suppress the function of plant TFs. The second of our examples presented here involved the use of the repression domain, SRDX, which consists of 12 amino acids with an improved strong repression domain capable of dominantly suppressing the function of plant TFs [34]. As in the RNAi example, to suppress CAG1 and CAG2 functions, we planned to use the 35S promoter and attached the SRDX to the C-terminus of CAG1 and CAG2 to suppress the function of these two types of CAGs, simultaneously. However, the use of the 35S promoter did not change the floral traits seen in RNAi transgenic chrysanthemum. Since the second intron of *AG* was available as a region to control the expression of class C genes in *Arabidopsis* [35], we planned to use the second intron as a promoter to express *CAG1-SRDX* and *CAG2-SRDX* genes instead of the 35S promoter. We isolated a second intron of *CAG1* and confirmed that it showed promoter activity via reporter GUS activity in both stamens and pistils in transgenic chrysanthemum [28]. Next, using the isolated second intron of *CAG1* as a promoter, chimeric repressors with SRDX for CAG1 and CAG2 were simultaneously introduced into chrysanthemum. This achieved petalized stamens and pistils in the transgenic chrysanthemum, as well as in RNAi transgenic chrysanthemum.

### 3.5. An Example of Improving Transgene Expression to Obtain Target Traits in Transgenic Chrysanthemum, Part 3

As another example, in 2017, truly blue chrysanthemum produced by genetic engineering technology was reported [27]. In this context, it had previously been reported that promoter selection was important [29] for generating blue chrysanthemum [27]. Regarding the gene transfer required for developing the blue chrysanthemum, the first step was to introduce the campanula *flavonoid 3′,5′-hydroxylase* gene (*CamF3′,5′H*), which resulted in the acquisition and accumulation of delphinidin-based anthocyanins to produce a new blue-purple flower color [29]. The *CamF3′,5′H* gene was initially expressed using the 35S promoter or the *chalcone synthase* gene (*CHS*) promoter, but this did not result in sufficient accumulation of delphinidin-based anthocyanins. Next, Noda et al. [29] used several promoters and different *F3′,5′H* genes derived from other plant species. Finally, the combination of the chrysanthemum *flavanone 3-hydroxylase* (*CmF3H*) promoter, *CamF3′,5′H*, and the 5′-untranslated region (UTR) of the tobacco (*Nicotiana tabacum*) *alcohol dehydrogenase* gene (*NtADH* 5′UTR) led to the highest delphinidin-based anthocyanin accumulation [29]. The truly blue chrysanthemum [27] that was eventually developed was based on the results of this previous study. The truly blue phenotype was generated by introducing two genes, *CamF3′,5′H* and *CtA3′5′GT* encoding butterfly pea (*Clitoria ternatea*) *UDP* (uridine diphosphate)-*glucose:anthocyanin 3′,5′-glucosyltransferase* [27]. These results show the importance of using a promoter appropriate for the target trait and the gene of interest (Figure 2).

It should be noted that the 35S promoter does function in chrysanthemum, with numerous examples of its use having been reported [22,36]. However, the general-purpose 35S promoter is not necessarily universally appropriate for all applications, so it is necessary to select an appropriate promoter for the particular purpose in chrysanthemum, as the above example shows (Figure 2).

## 4. Genome Editing in Chrysanthemum

### 4.1. Genome Editing in Horticultural Plants

In recent years, a number of genome-editing projects using TALENs and CRISPR/Cas9 have been reported in higher plants [37]. New genome-editing tools have also been developed, and with respect to the CRISPR/Cas system, options other than the CRISPR/Cas9 system are also becoming possible [38]. In terms of commercialization, in Japan, a genome-edited tomato, from which the genome-editing-related genes have been removed in the genome, with high GABA accumulation has been in commercial use since 2021 [37]. The GABA tomato (sanatechseed: https://sanatech-seed.com/en/work-en/ (accessed on 28 June 2023)) has been sold not only for processing but also as seedlings and can be eaten fresh. Ornamental plants have also been reported to have undergone a number of genome edits [39], but limited reports on chrysanthemum have been published. This might be explained by the characteristics of chrysanthemum that make genome editing difficult. Chrysanthemum is basically a hexaploid and self-incompatible. Due to its large genome, the whole-genome sequence has not been determined. Therefore, the confirmation of cleavage and mutagenesis of a large number of target genes is a major challenge in chrysanthemum compared with that in other plant species. For genome editing of chrysanthemum, the method is similar to genetic engineering because, at present, genome-editing vectors are genetically introduced into chrysanthemum by the *Agrobacterium* method. To introduce mutations in all target genes, it is necessary to create vectors with high cleavage efficiency for both CRISPR/Cas9 and TALENs.

### 4.2. Examples of Reported Genome Editing in Chrysanthemum

A few reports on genome editing in chrysanthemum have been published. One focused on the development of technology using transgenic chrysanthemum, in which the yellowish-green fluorescent protein of the marine plankton *Chiridius poppei* (*CpYGFP*) [40] was introduced as a foreign gene [41]. Using transgenic plants into which multiple copies of *CpYGFP* had been introduced, genome editing with the CRISPR/Cas9 system was performed on the genome-edited transgenic chrysanthemum into which the CRISPR/Cas9 vector had been introduced, as if the foreign gene introduced in multiple copies was an endogenous gene of chrysanthemum. It was reported that the accumulation of mutations can be promoted by obtaining lateral buds and performing repeated cuttings or by re-calcifying leaves of genome-edited transgenic plants to obtain shoots, using plantlets in which the vector is still introduced [41]. In this technique, repeated cuttings and re-calcifying are effective for the accumulation of mutations when mutations are not introduced in all target genes in a single genome-editing operation. Another genome-editing approach with TALENs [30] targeting the *DMC1* gene (conferring sterility traits), which controls homologous recombination occurring specifically during meiosis, has been reported. Although TALENs were used in that study, it is assumed that TALENs themselves are highly active against the *DMC1* gene. As chrysanthemum is a hexaploid species, it is assumed that it possesses multiple alleles for a single gene. These multiple alleles are also typically known to be highly heterozygous in chrysanthemum. Meanwhile, the *DMC1* gene has very high homology among multiple alleles, which was thought to be the key feature enabling a single TALEN vector to target all alleles at once.

### 4.3. Difficulties of Genome Editing in Chrysanthemum

When performing genome editing in chrysanthemum, extra care is needed with regard to various points, when compared with equivalent procedures performed in other plants. The first is the need to confirm the number of target genes and the conservation of the sequence. Chrysanthemum is basically a hexaploid, but the chromosome number may differ among varieties [6,7,8]. As a result, the number of target genes may differ from the expected number, so the number of target genes for each variety to be handled must be determined in advance. Then, if the obtained sequence information can reveal whether the gene is actually functional or not, it will be possible to determine whether it is necessary to introduce mutations into each allele. In addition, since chrysanthemum has a high degree of heterogeneity even among alleles of a particular gene, it is necessary to decode and sequence all alleles of a particular gene, as much as possible. For example, the *DMC1* gene, which is endogenous to chrysanthemum, for which genome editing has been reported, has a highly conserved sequence even among alleles within an individual, and its sequence is similar not only within chrysanthemum but also among all plant species [30]. Genes that are highly conserved among plant species such as this would aid further research on plants whose entire genomes have not been sequenced, including chrysanthemum.

The concept of target sequence selection for genome editing in chrysanthemum and the difficulties in selecting the target sequences compared with the case in other plants are explained here using the *CAG1* gene as an example. An alignment of six *CAG1* genes from the chrysanthemum cultivar “Sei-marine” [28] with a 240 bp coding region is shown in Figure 3. This alignment shows that the conserved regions between introns, the sequences from which the target sequence can be selected, are short. For these six genes, the PAM of SpCas9 [42] for the target sequences common to the six genes is shown, taking into account the introns (Figure 3; black arrows). This figure reveals that there are only a few options of target sequence from which to choose. In the case of chrysanthemum, it is recommended to select a target site that has a restriction enzyme site at the cleavage site for cleaved amplified polymorphic sequence (CAPS) analysis, if possible, and that shows a high cutting efficiency under such conditions. Although not seen in the alignment region in Figure 3, the heteroduplex mobility assay (HMA) is basically considered to be difficult in chrysanthemum because it is common among alleles of a gene for the number of bases to increase or decrease in multiples of 3. In addition, as mentioned above, because the number of chromosomes in chrysanthemum may differ among varieties, the number of genes in other varieties cannot be used as a reference, and it is necessary to confirm the number of target genes in the variety to be analyzed.

### 4.4. Proposal of Preliminary Trait Confirmation by RNAi for Genome Editing

Reports on previous studies on genome editing of other plant species have described that genome editing reproduced useful traits exhibited by mutants for which the causal gene had already been identified. When the causal gene and the mutant trait are identified, the effectiveness of the newly produced genome-edited vector can be confirmed. In addition, the produced vectors for genome editing should aid the accumulation of useful research results, such as the ability to deploy useful traits in many varieties. However, the acquisition of mutants and the analysis of causative genes are extremely difficult in chrysanthemum because of its high ploidy and large genome. Therefore, in chrysanthemum, it is difficult to reproduce mutant traits as in other diploid plants. In addition, it is assumed that the introduction of multiple mutations is required in the genome editing of chrysanthemum, and it is expected to take a long time to confirm the mutant trait after such genome editing. Although genome editing can be conducted by assuming the function of the target gene with reference to information on other plant species, it is desirable to confirm the trait obtained by mutation to the target gene in advance. Therefore, as an alternative to mutant analysis, the confirmation of target traits in advance by RNAi is considered reliable and efficient for genome-editing research in chrysanthemum. In fact, in chrysanthemum, after confirming the sterility trait in RNAi-*DMC1* transgenic chrysanthemum [43], the sterility trait was then reproduced by genome editing of the *DMC1* gene with TALENs [30]. We also have experience with genome editing in torenia from prior trait confirmation using RNAi. We generated RNAi transgenic torenias of two torenia (*Torenia fournieri* Lind.) class C genes, *PLENA* (*TfPLE*) and *FALINELLI* (*TfFAR*), and confirmed that their simultaneous suppression resulted in petalization of stamens and pistils [44]. Next, we confirmed that the simultaneous introduction of *TfPLE* and *TfFAR* mutations by CRISPR/Cas9 resulted in petalization of stamens and pistils, as in the RNAi transgenic torenia. Thus, the use of RNAi to confirm the trait change in advance would provide a reliable aid to decision-making about whether to proceed with mutagenesis by genome editing for the target gene.

### 4.5. Future Challenges for Increasing the Efficiency and Practical Application of Genome Editing in Chrysanthemum

Several options for improving the efficiency of genome editing match those for improving transgene expression (Table 1). For example, tools for efficient genome editing in chrysanthemum include the parsley *ubiquitin* (*PcUbi*) promoter, which achieves higher expression in the callus [26], the translation enhancer *NtADH* 5′UTR [22], and the highly efficient terminator HSPT878 [28,33]. A system for the transient ultra-high expression of proteins has been reported in tomato (Tsukuba system) [45]. To date, genome editing has been performed mainly by *Agrobacterium* methods; however, the genome-editing vectors or proteins are not introduced by the conventional *Agrobacterium* method. The following example involves a method for commercializing genome-edited mutants of crops obtained in the future to prevent genome-edited vectors from remaining in the genome. For example, bombardment using ribonucleoprotein against protoplasts has resulted in wheat genome-edited mutants without insertion of genome-edited genes [46]. Similarly, ribonucleoproteins were introduced into protoplasts by PEG-calcium transfection [47] of *Arabidopsis*, tobacco, lettuce, and rice, and DNA-free genome editing was performed [48]. However, the use of protoplasts may not be suitable for genome editing in chrysanthemum, as it appears to be difficult to reproduce traits of the parental lineage from protoplasts or for regenerated individuals. In potato, transient *Agrobacterium* infection of the TALEN vector resulted in genome-edited mutants in which the genome-editing vector did not remain in the genome [49]. The iPB (in planta particle bombardment) method is a bombardment method in which transgenes are launched directly into the stem apex site [50]. It has been shown that genome editing is possible in wheat by directly introducing not only transgenes but also ribonucleoproteins [51]. As another example, systems using hyperactive *piggyBac* transposase (hyPBase), in which genome-edited genes, once incorporated into the genome, are excised after mutagenesis by genome editing, have also been reported in rice [52]. Thus, the tools and methods that should be used will depend on the particular purpose for which they are employed. To date, few reports of genome editing in chrysanthemum have been published. This is due to the fact that the efficiency of genome editing in these plants has not yet been improved. It is expected that there is room for further development of vector sequences (Table 1) and methods for highly efficient genome editing for vegetatively propagated plants such as chrysanthemum.

## 5. Genome Analysis of Chrysanthemum in Recent Years

It has been difficult to analyze the genome of chrysanthemum for molecular breeding because of its complexity. Specifically, its genome is hexaploid, highly heterogeneous, and about 10 Gbp in size. However, next-generation sequencing (NGS) technology has helped to overcome these problems and reduced the cost of analysis. For example, when the wheat draft genome was published in 2014 [53], the genome sequence was fragmented into more than 10 M reads constituting 10 Gbp; that is, only 60% of the estimated genome size (17 Gbp). At that time, wheat genome sequencing was conducted with an Illumina platform while performing chromosome-by-chromosome sequencing. The quality of de novo genome assembly only by short-read NGS was comparable to the assembly by BAC-by-BAC sequencing with the 454 platform [54]. In 2018, an updated wheat genome sequence was composed of 21 chromosomes with 14.5 Gbp [55]. Assembly methods involved a combination of various sequencing technologies, for example, chromosome conformation capture (Hi-C) sequencing, Bionano optical mapping, and ChipSeq, along with various resources, such as BAC clones, radiation hybrid maps, and genetic maps. In particular, the assembly software package “DenovoMAGIC2TM (NRGene, Nes Ziona, Israel) made a major contribution to long and complex genome assembly. This process was upgraded and applied to multiple wheat genome assemblies [56].

To overcome the challenge of whole-genome sequencing, various target sequencing technologies were used. Restriction site-associated DNA (RAD)-Seq, genotyping by sequencing (GBS), and double-digest RAD (ddRAD)-Seq use restriction enzymes to condense the target genomic sequences [57,58,59]. Alternative technology was also developed to amplify the genome sequences through the use of random primers [60]. In *Chrysanthemum*, ddRAD-Seq was applied to construct genome-wide markers for the fine mapping study [61,62,63].

The development of technologies for longer sequencing also provided chromosome-level sequences. Genome assembly using PacBio Sequel II and the Oxford Nanopore sequencer in combination with Illumina was achieved even in non-model, polyploid, and heterogeneous plants, such as potato [64] and ground cherry [65]. For *Chrysanthemum*, two genome assemblies of *C. seticuspe* were released [66,67]. In the case of Gojo-0, a pure line bred from the self-compatible mutant of *C. seticuspe* (Maxim.) Hand.-Mazz, two sequence platforms, Illumina HiSeq 2500 and PacBio Sequel systems, were adopted, and Hi-C technology was used for the construction of a chromosome-level assembly from the assembled sequences. As a result, nine scaffolds corresponding to haploid chromosomes of Gojo-0 were generated These genome sequences provided us with ~70,000 annotated genes in *C. seticuspe*. Via comparative analysis, we can draw assumptions about the biological functions of genes from information on various domains and experimental reports from related species, such as *Arabidopsis thaliana*. We can also apply the genomic data to molecular breeding, even in hexaploid chrysanthemum.

## 6. Difficulty of the Application of Diversified Genome Sequences in Chrysanthemum

While the genome assembly of *C. seticuspe* shed light on the chrysanthemum genome-wide analysis, care must be taken in how this sequence information is used, for example, genome resequencing to construct the molecular markers based on the SNPs and small INDELs. Even if the differences of haplotypes and paralogs attributed to polyploidy are discarded, the reference genome sequences can help to discover random genomic differences among the compared varieties. Through the use of annotation data, we can speculate the orthologs from the combination of the reference genome sequences and transcriptome data of the targeted varieties. However, the use of a reference genome sequence is not suitable for statistical analysis because of the diverse sequences. For example, expression analysis is not recommended. When the chrysanthemum transcriptome data were mapped to the reference genome sequences, more than 20% of reads were unmapped (in preparation). This result indicated that the expression profiles could be affected by sequence conservation of the targeted genes, and the detection of differentially expressed genes from RNA-Seq would show erroneous candidates. If researchers focus on a certain gene, this effect can be ignored, because the effect of the sequence diversity is constant among the samples.

## 7. Transcriptome Analyses of Chrysanthemum in Recent Years

The automated annotation of genome sequences has enabled genes and their functions to be predicted. However, apart from performing experimental validation, it is difficult to confirm that the functional annotation is accurate. We have often used Gene Ontology (GO) terms to annotate genes, which are categorized into three groups: biological process, cellular component, and molecular function (http://geneontology.org/docs/ontology-documentation/ (accessed on 28 June 2023)). To assign GO terms, experimental validation should be conducted, while sequence homology is commonly used.

Detection of the expression of each gene in a certain condition is one of the most reliable methods to associate genes with biological phenomena. From PCR experiments for each gene to microarray analysis of listed genes, many strategies for expression analysis can be performed. NGS contributes to transcriptome analysis from two perspectives: expression profiling and gene structure determination. Since Illumina sequences provide many transcribed sequences, we mainly used this technology for expression analysis. Although we can predict gene structures using Illumina, we also recognize that there are many chimeric transcript isoforms because of the fragmented sequences. The best solution to the problem might be the usage of reference genome sequences. However, as we mentioned in the previous section, the lower mapping ratio by the sequence diversity largely affects the results. Meanwhile, longer NGS technologies have a major advantage for gene structure annotation since they enable full-length transcripts to be captured. In particular, longer NGS is a powerful way of annotating genes in polyploids because, in these cases, there are many paralogs, and we need to distinguish them using SNPs and exon–intron structures. Although we could obtain many isoforms from RNA-Seq by Illumina, we should also recognize many artifacts and chimeric structures in the data. In chrysanthemum, RNA-Seq analysis was performed for both purposes, expression analysis and comprehensive transcript construction [68,69], even if short- and long-read NGS technology could not solve the complexity of transcripts in chrysanthemum because of many heterogenous transcripts. In combination with the accumulating genomic information, transcriptome data will be a powerful tool for molecular breeding in *Chrysanthemum*.

## 8. Perspectives

There is still ample room for research and development in the field of chrysanthemum study. Chrysanthemum is a hexaploid and self-incompatible plant, making it almost impossible to obtain lines with uniform genomes. For propagation of the same variety, vegetative propagation by cuttings is performed, and the biological characteristics fundamentally differ from those of seed-fertile model plants, whose genome analysis has been progressing in recent years and whose molecular biology is becoming more deeply understood. In the future, whole-genome analysis of chrysanthemum may be advanced by the development of information-processing techniques on assemblages, as well as sequencing technology. Due to the rapid development of chrysanthemum cultivars and the large number of chrysanthemum varieties, research on target traits, mainly based on genome analysis, will accelerate in the future. The themes covered in this review (genetic engineering, genome editing, and genome analysis, including transcriptome analysis) are closely interrelated in the development of chrysanthemum research (Figure 4). In particular, the development of genome analysis will significantly contribute to the development of all research in genetic engineering, genome editing, and conventional breeding (colored arrowheads in Figure 4). Transcriptome analysis is a very useful source of information for regulating the expression of transgenes (brown arrowhead in Figure 4), especially for selecting a promoter for transgene expression, as well as for determining the target site (tissue and organs), specific timing, and stresses to which organisms are exposed, for the generation of transgenic chrysanthemum. To generate a mutant with the target trait by genome editing, it is effective to confirm the traits in advance using RNAi transgenic plants and/or mutants (purple arrowhead in Figure 4). Meanwhile, genome-edited mutants could become parental lines for crossbreeding (blue arrowhead in Figure 4). Genome analysis is also necessary to confirm the number of alleles of the target gene for mutagenesis by genome editing (brown arrowhead in Figure 4). Thus, each analysis (genetic engineering, genome editing, and genome analysis) provides a step toward achieving the goal, and the different methods and information obtained by them can be well-combined and used synergistically, instead of simply employing a single method.

Plants obtained through biotechnology and genomic breeding are often validated for their primary traits in artificial environments, such as plant growth chambers or closed greenhouses. On the other hand, for commercial applications, the developed horticultural crops are grown on a large scale, such as in the open field. The desired target traits that are confirmed in an artificial environment may not necessarily be reproduced in a large-scale cultivation environment. Therefore, as the second stage of trait confirmation, it is necessary to conduct cultivation tests several times under actual cultivation conditions, especially in the field for two to three years, as cultivation environments are highly variable. Rather than relying too much on biotechnology, this step-by-step process is considered important to maximize the use of biotechnology for future cultivation.

## 9. Conclusions

In chrysanthemum, many reports have been published on basic studies by genetic engineering or on the assignment of new traits, such as the generation of blue chrysanthemum. However, to achieve a desired trait, it is necessary to select appropriate expression tools, such as promoters, to correctly control the expression of the introduced gene. Although genome editing is becoming possible in chrysanthemum, it is not as widely used as in other plants due to this plant’s high ploidy (hexaploidy) and self-incompatible nature. Tools for efficient genome editing and further efficiency improvement in these plants are still needed. The use and further development of genome analysis are desirable for efficient and reliable implementation of genetic engineering and genome editing in chrysanthemum.

## Figures and Tables

**Figure 1 plants-12-02566-f001:**
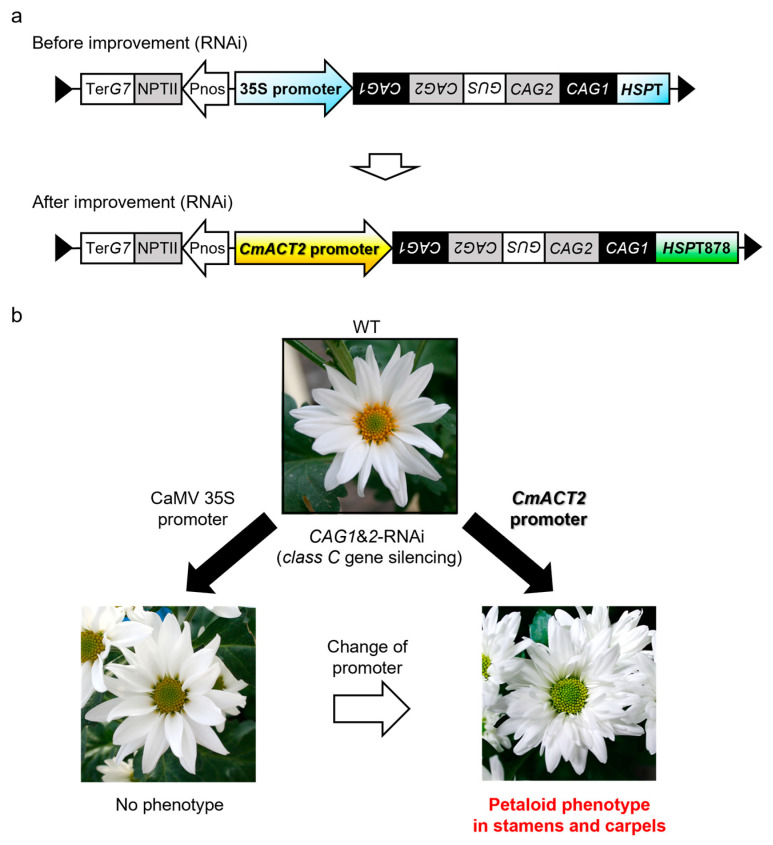
Improvements in class C gene suppression. (**a**) RNAi vector constructs before and after improvement. The 35S promoter and the *HSPT* (terminator) are changed into *CmACT2* promoter and *HSPT878*, respectively. (**b**) Change from 35S promoter to *CmACT2* promoter led to petalization of stamens and pistils in RNAi transgenic chrysanthemum plants. *GUS* was used as a spacer for the RNAi vector. WT; wild-type chrysanthemum (non-transgenic plant).

**Figure 2 plants-12-02566-f002:**
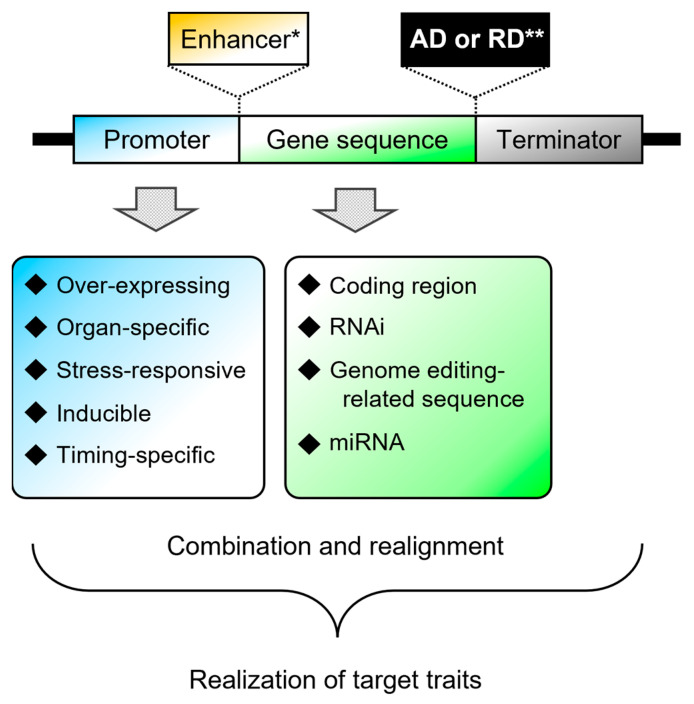
A combination of parts used to construct a vector to obtain the target trait in plants. * Optional. Used for increasing protein translation efficiency. ** Optional. Used for modification of the transcription factor function, such as dominant repression or activation of the transcription factor function. AD: activation domain, RD: repression domain.

**Figure 3 plants-12-02566-f003:**
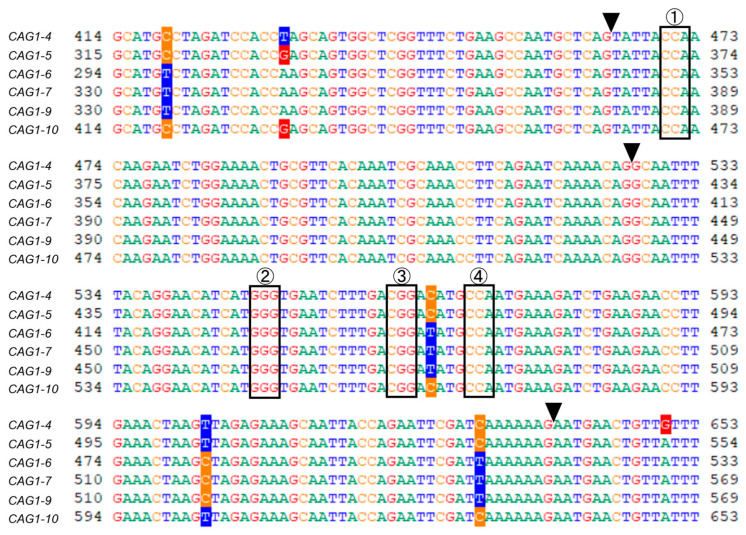
Alignment of part of the *CAG1* genes. Overall, 240 bp of the six *CAG1* genes [28] were aligned (GENETYX ver. 15; Genetyx Co. Ltd., Tokyo, Japan). PAM sequences for SpCas9 [42] are indicated by boxes (four PAMs are indicated with the numbers in circles). Black arrows indicate the position where the introns are inserted. There are four PAM sequences in the region of the aligned sequence.

**Figure 4 plants-12-02566-f004:**
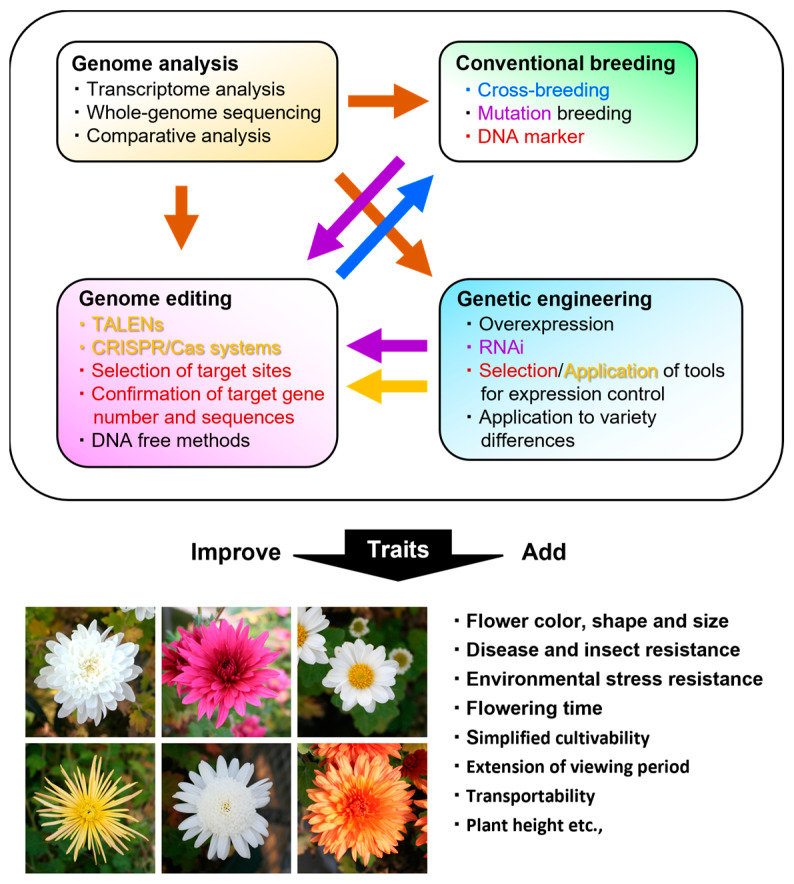
Illustration regarding the utilization or combination of experiments, such as genome analysis, conventional breeding, genetic engineering, and genome editing (upper) to achieve the desired trait(s) (lower). These old and new breeding approaches add or improve traits in chrysanthemum. For the frame at the top of the illustration, text and arrows of the same color correspond to each other.

**Table 1 plants-12-02566-t001:** Genetic information of tools for controlling transgene expression in plants.

Sequence Name	Source	Activity or Attribution	Analysis	Applied Plants	Ref.
*NtADH*-5′UTR	tobacco	translation efficiency (enhancer)	GUS activity	chrysanthemum	[22]
*AtADH*-5′UTR	Arabidopsis	translation efficiency (enhancer)	*cry1Ab*, *sarcotoxin IA*	chrysanthemum	[23]
Ω	TMV	translation efficiency (enhancer)	GUS activity	rice, tobacco	[24]
T*nos*	agrobacterium	transcription efficiency (terminator)	GUS activity	chrysanthemum	[20,25]
*HSP*T	Arabidopsis	transcription efficiency (terminator)	GUS activity	chrysanthemum	[26,27]
*HSP*T878	Arabidopsis	transcription efficiency (terminator)	*CAG*	chrysanthemum	[28]
*cab* promoter	chrysanthemum	promoter	GUS activity	chrysanthemum	[20]
*EF1a* promoter	tobacco	promoter	GUS activity	chrysanthemum	[25]
*F3H* promoter	chrysanthemum	promoter (petal)	*F3′5′H*, *A3′5′GT*	chrysanthemum	[27,29]
CaMV 35S promoter	CaMV	promoter	GUS activity	chrysanthemum	[20,22,25,26]
*CmACT2* promoter	chrysanthemum	promoter	GUS activity, *CAG*	chrysanthemum	[26,28]
*PcUbi* promoter	parsley	promoter	GUS activity	chrysanthemum	[26]
*CAG* 2nd intron	chrysanthemum	promoter (stamen, carpel)	GUS activity, *CAG*	chrysanthemum	[28]
P*mas* 201	agrobacterium	promoter (dual)	*CmDMC1*	chrysanthemum	[30]
SRDX	artificial	RD for transcription factor	*CAG*	chrysanthemum	[28]
transactivation domain of VP16	herpes simplex virus	AD for transcription factor	*FLC*	Arabidopsis	[31]

## Data Availability

The data presented here are available in all the publications cited in this review.

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
