# Peer review of "Overcoming Difficulties in Molecular Biological Analysis through a Combination of Genetic Engineering, Genome Editing, and Genome Analysis in Hexaploid Chrysanthemum morifolium"

_plants, 2023, doi:10.3390/plants12132566_

Round 1
Reviewer 1 Report
In the manuscript, the authors discussed the use of biotechnology in chrysanthemum, the development of genomic information on these plants, and the future prospects for using biotechnology in chrysanthemum. However, to my knowledge, each of these issues has been reported in previous review articles or reviewed recently. According to the title and perspectives in the manuscript, in my understanding, the aim of this review is addressing the problems on molecular analysis of key traits or molecular breeding in chrysanthemum by integrating the biotechnological tools and genomic tools, which is an interesting topic. But, the current structure and content of this manuscript can’t achieve the goals that the authors want to express. In model plants or in important crops, there is a conventional way to solve them. First, genomic tools such as genome resequencing, GBS, RNA-Seq, genetic maps, GWAS and so on are used to figure out the genes of gene locus responsible for the trait variations among varieties. Then, biotechnological tools such as transgenic overexpression, RNAi and genome editing are used to verify the gene-trait relations. Is this way also applicable to chrysanthemum? How many gene-trait relations are decoded by biotechnological tools or by genomic tools or by both? These questions should be answered comprehensively and systematically. In a word, the manuscript should be rewritten in a more reasonable form to summarize the history and progress in molecular researches in chrysanthemum.
Author Response
>Our response
Thank you very much for your valuable comments.
We have tried to differentiate our review from similar reviews in the past by introducing some of the challenges we have tried in the past, from failure to success (mainly section 3-2, 3-3, 3-4). In addition, based on your comments, we added descriptions for genomic tools. First, genome resequencing and GBS including other target resequencing technologies were describe in the section 5 (lines 397-403). In this paragraph, we added references for genetic map analysis ([64-67]). We also added a new section 6 “Difficulty of the application of diversified genome sequences in chrysanthemum” to explain what kinds of problems exist to use NGS data for chrysanthemum analysis and how to utilize the data. Even though I showed unpublished data, one manuscript relating to the data has been submitted to a journal. Second, since transcriptome analysis using RNA-Seq has been described in the previous section 6, we just added the following sentences in section 7 (lines 455-457).
“The best solution of the problem might be the usage of reference genome sequences. However, as we mentioned in the previous section, the lower mapping ratio by the sequence diversity largely affect to the results.”
  In addition, as pointed out by two other reviewers, we have added new texts on cross-breeding (lines 73-86) and on the prospects for commercial use of plants obtained through biotechnology and genome breeding (lines 500-509), which have been poorly described in the past.
  These sentences are written in red letters. We hope you will find them.

Reviewer 2 Report
Chrysanthemum is an important floriculture plant with immense economic importance. The present review discusses the genetic engineering, genome editing, genome and transcriptomic approaches in this economically important crop. The paper is well written and provides novel information not covered in any other review. I suggest somewhat broadening of the section on conventional breeding approaches which in my opinion has been scantly dealt with. I recommend the acceptance of the article after this minor change.
Author Response
Reviewer 2
Chrysanthemum is an important floriculture plant with immense economic importance. The present review discusses the genetic engineering, genome editing, genome and transcriptomic approaches in this economically important crop. The paper is well written and provides novel information not covered in any other review. I suggest somewhat broadening of the section on conventional breeding approaches which in my opinion has been scantly dealt with. I recommend the acceptance of the article after this minor change.
>our response
Thank you very much for your appreciation of our review.
We have followed your suggestion and added a new sentence in red (lines 73-86) regarding crossbreeding. In addition, according to other reviewers, we added the sentences. Those sentences are written in red letters. We would appreciate it if you could check it.

Reviewer 3 Report
In my opinion it is an excellent manuscript with practical points of view robustly based on the experience acquired in the improvement of plants, in this case the chrysanthemums.
I would only recommend including some mention (with references) about the need to relate any type of biotechnological tool with traditional breeding, for example the need to evaluate any type of modification under field conditions, that is, the genotype-environment interaction. In this sense, it is necessary to evaluate the elite clones in different locations for at least 2-3 consecutive years. The fact of avoiding these last stages has originated, in my opinion at the end of the days, a false expectation about the efficiency of the improvement of plants (specifically polyploids) assisted by biotechnologies, being prevailing to recover this perception
Author Response
Reviewer 3
In my opinion it is an excellent manuscript with practical points of view robustly based on the experience acquired in the improvement of plants, in this case the chrysanthemums.
I would only recommend including some mention (with references) about the need to relate any type of biotechnological tool with traditional breeding, for example the need to evaluate any type of modification under field conditions, that is, the genotype-environment interaction. In this sense, it is necessary to evaluate the elite clones in different locations for at least 2-3 consecutive years. The fact of avoiding these last stages has originated, in my opinion at the end of the days, a false expectation about the efficiency of the improvement of plants (specifically polyploids) assisted by biotechnologies, being prevailing to recover this perception
>our response
Thank you very much for your high appreciation of our review. In accordance with your suggestion, we added a new sentence in red regarding the precautions for the use of plants obtained through biotechnology and genome breeding (lines 492-502). In addition, according to other reviewers, we added the sentences. Those sentences are written in red letters. We would appreciate it if you could check it. We would appreciate it if you could check it.
